# The Fibrinogen-like Domain of ANGPTL3 Facilitates Lipolysis in 3T3-L1 Cells by Activating the Intracellular Erk Pathway

**DOI:** 10.3390/biom12040585

**Published:** 2022-04-16

**Authors:** Simone Bini, Valeria Pecce, Alessia Di Costanzo, Luca Polito, Ameneh Ghadiri, Ilenia Minicocci, Federica Tambaro, Stella Covino, Marcello Arca, Laura D’Erasmo

**Affiliations:** Department of Translational and Precision Medicine, Policlinico Umberto I, Sapienza University of Rome, viale dell’Università n. 37, 00161 Rome, Italy; simone.bini@uniroma1.it (S.B.); valeria.pecce@uniroma1.it (V.P.); lucapolitoo@gmail.com (L.P.); ghadiria64@gmail.com (A.G.); ilenia.minicocci@uniroma1.it (I.M.); federica.tambaro@uniroma1.it (F.T.); stella.covino@uniroma1.it (S.C.); marcello.arca@uniroma1.it (M.A.)

**Keywords:** ANGPTL3, lipolysis, lipid metabolism, adipose tissue, fibrinogen-like domain, ERK pathway

## Abstract

Background: ANGPTL3 stimulates lipolysis in adipocytes, but the underlying molecular mechanism is yet unknown. The C-terminal fibrinogen-like domain of ANGPTL3 (ANGPTL3-Fld) activates the AKT pathway in endothelial cells. Hence, we evaluated whether ANGPTL3-Fld stimulates lipolysis in adipocytes through the MAPK kinase pathway. Materials and Methods: 3T3-L1 adipocytes were treated with isoproterenol (ISO), ANGPTL3-Fld, or both. Lipolysis was evaluated through the release of free fatty acids (FFAs) in the culture medium. The activation status of intracellular kinases was evaluated with and without the inhibition of the BRAF–ERK arm of the MAPK pathway. Results: ANGPTL3-Fld alone was not able to activate lipolysis, while the combination of ANGPTL3-Fld and ISO determined a 10-fold enrichment of the FFA concentration in the culture medium with an incremental effect (twofold) when compared with ISO alone. ANGPTL3-Fld alone inhibited hormone-sensitive lipase (HSL), whereas the treatment with ISO induced the activation of HSL. The net balance of ANGPTL3-Fld and ISO cotreatment resulted in HSL activation. The results indicate that ANGPTL3-Fld generated an intracellular activation signal involving the MAPK–ERK pathway, possibly through the PDGFRβ—PLCγ-AMPK axis. Conclusion: ANGPTL3-Fld appears to act as a facilitator of lipolysis in adipocytes, and this effect was driven by a signal mediated by a pathway that is different from the canonical β-adrenergic stimulus.

## 1. Introduction

Angiopoietin-like 3 (ANGPTL3) is a key player in lipid and lipoprotein metabolism due to its ability to inhibit both the lipoprotein (LPL) and endothelial lipases (EL) [1,2]. Consistently, it was observed that the overexpression of ANGPTL3 in mice causes hypertriglyceridemia [3], whereas the loss-of-function mutation of ANGPTL3 in humans causes familial combined hypolipidemia (FHBL2, OMIM #605019). FHBL2 has been reported as being associated with an increased LPL activity, a low fasting concentration of plasma triglycerides (TG), and an increased removal of triglyceride-rich lipoproteins (TRL) during the postprandial phase [4,5]. At the molecular level, ANGPTL3 seems to be involved in the regulation of TRL metabolism. Indeed, several lines of evidence indicate that this protein is also involved in the control of adipose tissue physiology and substrate trafficking [1]. In particular, Shimamura et al. [6] reported that ANGPTL3 is able to stimulate the release of FFA in 3T3-L1 adipocytes. The finding that an ANGPTL3 deficiency in humans is associated with reduced levels of circulating FFA appears to support this model [7,8].

The mechanism underlying the effect of ANGPTL3 on lipolysis in adipose tissue remains unclear. Camenisch et al. [9] investigated the binding of ANGPTL3 to endothelial cells and observed that the C-terminal, the fibrinogen-like domain of this protein (ANGPTL3-Fld), is recognized by the integrin receptor α_ν_β_3_ and induces the activation of the serine/threonine protein kinase (AKT). Among the main mitogen-activated protein kinase (MAP kinases), the extracellular signal-regulated kinase (ERK) and AKT represent the point of junction of the activation of several intracellular pathways [10,11]. It is well known that protein kinase A (PKA) and AMP-activated protein kinase (AMPK) are directly involved in the adipose tissue lipolysis mediated by the β-adrenergic stimulus [12]. The hormone-sensitive lipase (HSL), the major effector of the β-adrenergic-dependent lipolysis, is the downstream target of PKA and AMPK [12].

According to the accepted model [1,2], the N-terminal portion is mainly responsible for LPL inhibition [6], whereas the function of the ANGPTL3-Fld has been partially understood. Therefore, it is possible to postulate that the prolipolytic action of ANGPTL3 on adipocytes may reside in its Fld portion, and this could involve the activation of the MAPK pathway.

The main aim of the present study was to investigate whether ANGPTL3-Fld is able to stimulate lipolysis in adipose tissue and, eventually, to study the intracellular pathways involved in the transmission of the lipolytic signal.

## 2. Material and Methods

The 3T3-L1 cell line after differentiation was used as an adipocyte model. Cells were treated with Isoproterenol (ISO), ANGPTL3-Fld, or a combination of both. ISO was used as the standard β-adrenergic lipolysis mediator. Lipolysis was measured in the medium through the quantification of free fatty acids (FFAs), and the concentration increased per minute. Western blot analysis was used to test the main MAPK activation status (ERK and AKT), the phosphorylation status in the mediators of the β-adrenergic signaling (PKA and AMPK) and HSL, and the final effector of lipolysis. The phospho-kinase array analysis allowed us to deeply study the activated pathways in the different experimental conditions.

To verify that the activation of ERK was mediated by ANGPTL3-Fld stimulation, we inhibited the BRAF–ERK arm using vemurafenib [13]. Kinases that were not involved in the signal transmission showed a nonsignificant change. The kinases upstream of the inhibited target showed higher phosphorylation levels, while those downstream had lower ones.

### 2.1. Cell Culture and Treatments

3T3-L1 cells were purchased from the American Type Culture Collection (ATCC). Cells were cultured in Dulbecco’s Modified Eagle Medium (DMEM (Gibco-BRL Division, Thermo Fisher Scientific, Waltham, MA, USA)) containing 10% newborn bovine serum (NBS), 1% sodium pyruvate, 100 U/mL of penicillin, and 100 ug/mL of streptomycin (Gibco-BRL Division, Thermo Fisher Scientific, Waltham, MA, USA) incubated at 37 °C in an atmosphere of 5% CO_2_. 3T3-L1 cell differentiation into adipocytes was induced with DMEM containing fetal bovine serum (FBS), 100 U/mL of penicillin, and 100 ug/mL of streptomycin (DMEM–FBS–PEN–STREP) added with 1 μM dexamethasone, 500 μM isobutylmethylxanthine (IBMX), and 1 μg/mL insulin (Thermo Fisher Scientific, Waltham, MA, USA). After 48 h, the medium was replaced with the DMEM–FBS–pen–strep added with 1 μg/mL of insulin. The differentiation status was maintained using DMEM–FBS–PEN–STREP; the medium was replaced every 48 h. Cells were cultured for a total of 14 days. The adipocytes differentiation was assessed using the red-oil stain, as reported in Appendix A.

After differentiation, 3T3-L1 adipocytes were treated with 100 nM of isoproterenol (ISO) (Thermo Fisher Scientific, Waltham, MA, USA) or 100 nM of recombinant human ANGPTL3-Fld (AdipoGen Life sciences, San Diego, CA, USA) for 30 min, and the cotreatment was performed with ANGPTL3-Fld for 10 min followed by the addition of ISO for 30 min. Untreated 3T3-L1 adipocyte cells were used as the negative control. Therefore, three different conditions were tested in all the experiments: ISO alone, ANGPTL3-Fld alone, and the combined treatment with ISO and ANGPTL3-Fld. Isoproterenol is a β-adrenergic receptor agonist and has been demostrated to stimulate in vitro loipolysis on 3T3-L1 adipocytes [14].

### 2.2. Quantification of Free Fatty Acid Concentrations

Free fatty acids (FFAs) were quantified in culture media using the lipolysis kit provided by ZenBio (#LIP-2-NC). The assay was performed according to the manufacturer’s instructions.

Briefly, the stimulation of the lipolysis led to the release of a quinoneimine dye that showed maximum absorbance at 540 nm. The increase in absorbance at 540 nm was directly proportional to the free fatty acid (FFA) concentration in the sample. The FFA concentration increase in the medium was analyzed using the Formula (1) and expressed as μM/min.
(1)$$\frac{\mathrm{FFA}\;\mathrm{concentration}\;60\;\min—\mathrm{FFA}\;\mathrm{concentration}\;30\;\min}{30}$$

### 2.3. Western Blot Analysis

Total proteins were extracted from 3T3-L1 adipocytes by using a 2% beta-mercaptoethanol RIPA Lysis Buffer (20 mM Tris-HCl pH 8.0, 400 mM NaCl, 5 mM EDTA, 1 mM EGTA, 1 mM Na pyrophosphate, 1% Triton X-100, and 10% glycerol) supplemented with protease inhibitors (Roche, Indianapolis, IN, USA) and phosphatase inhibitors (Sigma, Saint-Louis, MO, USA). Total protein concentration was determined using the bicinchoninic acid (BCA) Protein Assay Reagent. Proteins were fractionated by SDS-PAGE (8% and 10%) and transferred to a nitrocellulose membrane according to the manufacturer’s protocols (Bio-Rad, Hercules, CA, USA). The membranes were incubated with primary antibodies against HSL (1:1500) (Cell signaling), pHSL Ser565 (1:2000) (Cell signaling), pHSL Ser660 (1:2000) (Cell signaling), ERK1/2 (1:2000) (Cell signaling), pERK1/2 (1:2000) (Cell signaling), AMPK-a (1:1000) (Cell signaling), pAMPK-a (1:1000) (Cell signaling), AKT (1:2000) (Cell Signaling), pAKT (1:1500) (Cell signaling), PKA-c (1:2000) (Cell signaling), pPKA-c (1:2000) (Cell signaling), and GAPDH (1:3000) (Cell signaling) at 4 °C for 12 h. Membranes were then incubated with a 1:10,000 dilution of peroxidase-conjugated antirabbit antibodies for 2 h. Blots were developed with the ECL system. The chemiluminescence was acquired with the ChemiDoc MP Imager system (Bio-Rad). The densitometric analysis was performed using the Image Lab Software (Bio-Rad). For each band, the pixel intensity was analyzed. Total protein pixel intensity was normalized for those of the GAPDH, and the pixel intensity of phosphorylated forms was normalized for those of the total protein.

### 2.4. Human Phospho-Kinase Array

The detection of the phosphorylation status of 39 human kinases was determined with the Proteome Profiler membrane-based immunoassay “Human Phospho-Kinase Array Kit” (pARRAY, R&D Systems, Minneapolis, MN, USA). Total proteins were extracted from 3T3-L1 adipocytes and treated with 10 μM of vemurafenib or DMSO as control conditions for 24 h. The control or treated cells were stimulated with ISO or ANGPTL3-fld as described above. Vemurafenib selectively binds to the ATP-binding site of BRAF kinase and inhibits its activity resulting in an inhibition of the ERK arm of the MAPK pathway downstream [13]. As reported by Pecce et al. [15], 100 μg of total proteins were assayed with the Human Phospho-Kinase according to the manufacturer’s instructions [15]. The acquisition of the chemiluminescent signal of the membrane of the array was performed using ChemiDoc MP Imager (Bio-Rad). The analysis was done by using Image Lab Software (Bio-Rad) with the volume tool analysis methods. A 1.5-fold increase of phosphorylation level was set as the cut-off value for effective activation.

### 2.5. Statistical Analysis

All the data are reported as mean ± the standard deviation of three independent experiments. Statistical analyses were performed by unpaired t-tests. Differences were considered statistically significant with a *p*-value less than 0.05.

## 3. Results

### 3.1. ANGPTL3-Fld Enhances the Lipolysis in 3T3-L1 Cells Treated with Isoproterenol

The lipolysis in 3T3-L1 was firstly evaluated as the increase of the FFA concentration per minute in the growth medium after treatment with ISO (to evaluate the β-adrenergic-dependent lipolysis), recombinant human ANGPTL3-Fld, or a combination of both.

When compared with untreated cells, the treatment with ISO determined a fivefold increase in the FFA concentration in the medium, while no appreciable changes were observed after treatment with ANGPTL3-Fld alone (Figure 1). However, when 3T3-L1 adipocytes were treated with both ISO and ANGPTL3-Fld, we found a 10-fold increase (*p* < 0.01) in the FFA concentration in the medium. These findings indicate that the presence of ANGPTL3 amplifies lipolysis when a β-adrenergic stimulus is present.

### 3.2. ANGPTL3-Fld Facilitates Lipolysis Mainly Activating the ERK Arm of the MAPK Pathway

To investigate the mechanisms underlying the effect of ANGPTL3-Fld in sensitizing cells to the β-adrenergic stimulus, we firstly analyzed the activation of the two arms of the MAPK pathway, BRAF–ERK and PI3K–AKT, in adipocytes treated with ISO, ANGPTL3-Fld, or a combination of both (Figure 2) [12]. Compared with the untreated 3T3-L1 adipocytes, the phosphorylation level of AKT did not change (Figure 2A), whereas the one of ERK appeared to be increased in all experimental conditions (Figure 2B).

Subsequently, we investigated the activity of kinases typically involved in the β-adrenergic signaling: PKA and AMPK (Figure 3). We observed that the treatment with ISO displayed the highest levels of phosphorylated PKA, about a fivefold increase as compared with untreated cells (*p* < 0.01). In addition, the ANGPTL3-Fld treatment activates PKA, showing a threefold increase as compared with control cells (*p* < 0.01). All treatments determined an increase in phosphorylation levels of AMPK (pAMPK) that was lower in the case of ISO alone. Both ANGPTL3-Fld alone and the combined treatment with isoproterenol resulted in a significant increase in pAMPK (*p* < 0.05), the latter one leading to the maximal effect (threefold increase) when compared to untreated cells (Figure 3A).

Then, we analyzed changes in the phosphorylation levels of HSL, which represent the final effector of all prolipolytic signals [12]. In this regard, we examined the phosphorylation status of the two serine sites that activate (Ser660) or inhibit (Ser565) the activity of HSL, respectively. As compared with basal conditions, we observed a threefold increase in phosphorylation levels of the inhibitory residue when the cells were treated with ANGPTL3-Fld (*p* < 0.01). It is noteworthy that in the same experimental conditions, a slight increase in phosphorylation at the Ser660 residue of HSL was also observed. Conversely, when cells were treated with ISO or a cotreatment, we observed higher levels of phosphorylation of Ser660 that seem to be in favor of the stimulation in enzyme activity (Figure 3B).

These findings suggest that the signal transduction after the ANGPTL3-Fld treatment starts with the activation of the BRAF–ERK arm of the MAPK pathway. However, the observation that the phosphorylation balance was in favor of the inhibitory serine Ser565 residue led us to hypothesize that ANGPTL3-Fld alone may exert a net inhibitory activity on the late step of the lipolytic cascade. This agrees with the failure to induce a significant increase in the release of FFA when adipocytes are treated with ANGPTL3-Fld alone.

### 3.3. ANGPTL3-Fld Activates the PDGFRβ Receptor

To further understand which pathways are involved in the prolipolytic effect observed in cotreated adipocytes, we investigated the overall activation status of intracellular kinases after the treatment with ISO, ANGPTL3-Fld, or a combination of both. To this end, we determined the phosphorylation status of 39 intracellular kinases using a commercial phospho-kinase array.

As reported in Figure 4A, we observed that each treatment is able to determine a different kinase activation profile. In particular, the treatment with ISO activated the largest number of intracellular kinases. It activates PLCγ and three out of five second messengers of the Src family (the nonreceptor tyrosine kinases that transfer the signal inside the cells, SRC, FGR, and YES) [16] and kinases of the MAPK pathway as well as the STAT1, STAT3, and STAT6 proteins, which are signal transducers and transcription factors involved in energy metabolism [17]. Conversely, the treatment with ANGPTL3-Fld activates mainly the BRAF–ERK arm of the MAPK pathway (with the activation of CREB and PYK2, the endothelial nitric oxide synthase (eNOS), the STAT3 and STAT6, and the Heat Shock Proteins (HSP60 and HSP27). Therefore, as observed after ISO treatment, ANGPTL3-Fld-treated cells also displayed increased phosphorylation of several kinases involved in the energy metabolism [17,18,19]. In particular, the signal transduction of ANGPTL3-Fld-treated cells starts with the activation of eNOS, a well-known target of AMPK in the energy metabolism [17,18,19]. Finally, the cotreatment activated the smallest number of kinases such as the YES kinase, RSK1 in the serine 380, and STAT3 in the Y727 [17].

Altogether, our results show an involvement of the BRAF–ERK pathway during all treatments with the activation of downstream kinases involved in energy metabolism. To demonstrate that there is a specific involvement of the BRAF–ERK arm, we treated 3T3-L1 adipocytes with Vemurafenib. This molecule is a specific inhibitor of BRAF, hence inhibiting the BRAF-mediated activation of the ERK arm of MAPK [10,11]. After 48 h treatment with Vemurafenib, we repeated the phospho-kinase array analysis in cells stimulated with ISO, ANGPTL3-Fld, or a combination of both (Figure 4B). As expected, after the BRAF inhibition by Vemurafenib, the level of phosphorylation of ERK decreased. After inhibition with Vemurafenib, cells treated with ISO did not show the activation of any pathways further demonstrating the specificity of the BRAF–ERK arm of MAPK in the signal transduction of the lipolytic stimulus. However, when the Vemurafenib-inhibited adipocytes were treated with ANGPTL3-Fld, a different profile of kinase stimulation was noted. In these conditions, cells showed an intensification of the signal upstream of the BRAF–ERK arm that started with PDGFRβ and then propagated through PLCγ, FGR, and YES as second messengers; among these, PLCγ exhibited the higher activation (Figure 4B). This signal led downstream to the activation of p38 and STAT2.

When adipocytes were co-stimulated with ANGPTL3-Fld and ISO in the presence of BRAF–ERK inhibition, there was a switch of signal in favor of AKT, phosphorylated in both the Ser473 and Thr308 activator sites. Since crosstalk between the AKT and the ERK arm of the MAPK pathway is well-known [10,11], the finding of AKT activation confirms that the ERK pathway is involved in the role of facilitator of lipolysis as demonstrated by ANGLPTL3-Fld.

Finally, the results from the phospho-kinase array (Figure 4) strongly suggest that the action of ANGPTL3-Fld on adipocytes is potentially mediated by PDGFRβ and PLCγ as its second messenger.

## 4. Discussion

The elucidation of the direct effect of ANGPTL3 on adipocytes represents an area of growing interest due to accumulating evidence suggesting that this protein may have a relevant role in the regulation of FFA turnover and partitioning [1,2].

As the N-terminal domain of ANGPTL3 is involved in the lipases inhibitory activity of this protein, we searched for a possible role of its fibrinogen-like domain (Fld) on adipocytes. We showed that the ANGPTL3-Fld was not able to directly stimulate the release of FFA, but it amplified the lipolytic effect induced by the β-adrenergic stimulation. This finding is apparently in contrast with the one reported by Shimamura et al. [6] who showed that ANGPTL3 stimulated the FFA release in cultured adipocytes. This might be dependent on the differences in the experimental conditions. Because Shimamura et al. [6] used the full-length protein for their experiment, it is possible that the N-terminal domain of ANGPTL3 (17–207 residues) also carries a pro-lipolytic signal. However, this seems unlikely since it associates with ANGPTL8 in vivo, and the complex appears to be fully engaged in the LPL inhibition activity [20,21]. Alternatively, it is possible that the presence of the N-terminal domain produces some conformational change that is able to determine a complete pro-lipolytic activity of ANGPTL3. Further investigations are needed to clarify these aspects.

We found that the treatment of adipocytes with ANGPTL3-Fld was able to produce a marked amplification of the lipolytic effect induced by the β-adrenergic stimulation. While the β-adrenergic stimulus activated PKA and then AMPK thus inducing downstream HSL activation and lipolysis, the treatment with ANGPTL3-Fld alone similarly stimulated AMPK but instead generated an inhibitory effect on HSL. Interestingly, the activation levels of PKA and AMPK were opposite among 3T3-L1 cells treated with ISO vs. cotreatment. The treatment with ANGPTL3-Fld appears to activate both using a yet unknown molecular pathway that is presumably downstream of ERK. While PKA is commonly activated by the increase of intracellular cAMP after the activation of adenylate cyclase, AMPK is activated and silenced by many intracellular pathways; only some of them comprise the same that could activate PKA. Hence, in the case of a cotreatment, the different behavior of the two kinases could be explained by the contemporary activation of the ERK pathway mediated by ANGPTL3-Fld.

Since the effect of ANGPLT3-Fld treatment alone on HSL was inhibitory, we searched for other possible alternative pathways through which ANGPLT3-Fld may exert its pro-lipolytic effect. We found that all treatments (ANGPTL3-Fld, ISO, and the cotreatment) converged downstream on the activation of STAT3, which is a known transcription factor for genes involved in lipolysis and beta oxidation [22,23]. Based on this evidence, we can speculate that ANGPTL3-Fld induces in adipocytes the activation of one intracellular pathway involved in lipolysis. Both ISO and ANGPTL3-Fld used the BRAF–ERK arm of the MAPK pathway to transduce the lipolytic signal, and ANGPTL3-Fld amplifies the β-adrenergic lipolysis by an alternative mechanism that does not include the activation of HSL. The combined treatment using the two molecules one after the other avoided competition on the same targets allowing us to observe an additive effect on lipolysis.

The specific inhibition of the BRAF–ERK arm through vemurafenib highlighted that the treatment with ANGPTL3-Fld activated the platelet-derived growth factor receptor β (PDGFRβ). PDGFRβ is another tyrosine-activated protein engaging several well-characterized signaling pathways [24,25]. In addition, PDGF-β/PDGFRβ signaling plays a unique and essential role in the initiation of adipose tissue angiogenesis responding to tissue expansion during the development of obesity [24,25]. We theorize that ANGPTL3-FLd might be able to modulate lipolysis by its effect on PDGFRβ, which in turn activates the BRAF–ERK arm of the MAPK pathway and then, downstream, the STAT proteins (Figure 5). Whether PDGFRβ may represent a specific receptor for ANGPTL3-Fld is an attractive hypothesis. The treatment with vemurafenib might produce an increase in EGFR activation as rapid feedback, but not in PDGFRβ [13], thus sustaining a specific ANGPTL3-mediated activation that requires further and more direct demonstrations.

We would like to put our results in the context of the current model describing the role of ANGPTL3 in regulating the flux of energy substrate by partitioning fatty acids and triglycerides between white adipose tissue (WAT) and oxidative tissues [2]. We can speculate that ANGPTL3-Fld inhibits HSL during feeding thus signaling the presence of energetic substrates and acting as a feeding sensor. Conversely, in conditions such as intense exercise, cold, or starvation it contributes to lipolysis by preparing adipocytes for the β-adrenergic stimulation, thereby, increasing the availability of FFA [1,26]. A better understanding of the molecular cross talking between ANGPTL3 and other intracellular pathways, as well as the significance of its interactions with other members of its own family, such as ANGPTL4 and ANGPTL8, could be crucial to shed light on mechanisms underlying lipid and energetic metabolism related to the ANGPTL3 action.

In interpreting our findings, some study limitations should be considered. First, we did not compare the potential pro-lipolytic action of the N-terminal domain of ANGPTL3. Further studies are needed to clarify the dynamic effects of ANGPTLs on adipose tissue and energy substrate trafficking. Moreover, we carried out the experiments after a short period of adipocyte stimulation with ANGPTL3-Fld, thus preventing us from verifying what happens during a chronic stimulation period. However, it should be remembered that in conditions of a genetically determined absence of pharmacological inhibition of circulating ANGPTL3 by monoclonal antibodies, the level of FFAs is reduced in vivo [8,27]. The demonstration that in these latter conditions the β-adrenergic stimulation of lipolysis is blunted might represent the definite demonstration of our hypothesis. To this end, additional studies are strongly recommended.

## 5. Conclusions

In conclusion, ANGPTL3-Fld appears to stimulate the PDGFRβ—ERK—AMPK—STAT3 cascade thus enhancing the β-adrenergic-induced lipolysis. This effect is driven by the activation of a transduction signal that is different from the one passing throughout the canonical β-adrenergic pathway.

## Figures and Tables

**Figure 1 biomolecules-12-00585-f001:**
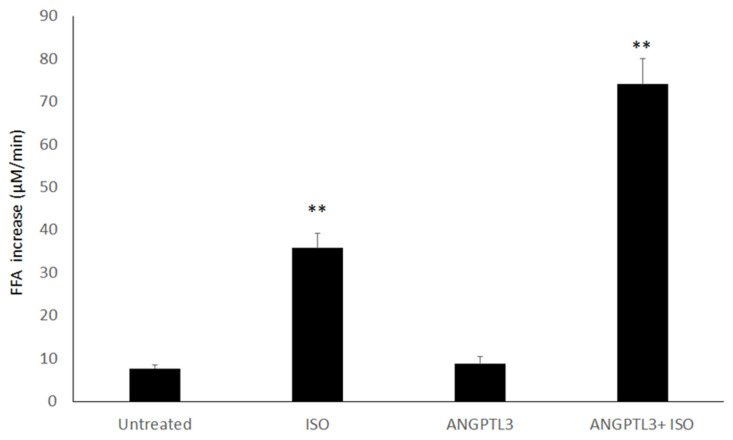
ANGPTL3 pretreatment promotes lipolytic effects in adipocytes. Free fatty acid (FFA) concentration increase (μM/min) in 3T3-L1 cell line treated for 30 and 60 min with ISO, recombinant human ANGPTL3, or pretreated for 10 min with ANGPTL3, and treated for 30 and 60 min with ISO (ANGPTL3 + Iso). The samples were compared with 3T3-L1 untreated cells. Data were expressed as mean ± SD of three biological replicates of the experiment. *p* value: ** < 0.001 (*t*-test).

**Figure 2 biomolecules-12-00585-f002:**
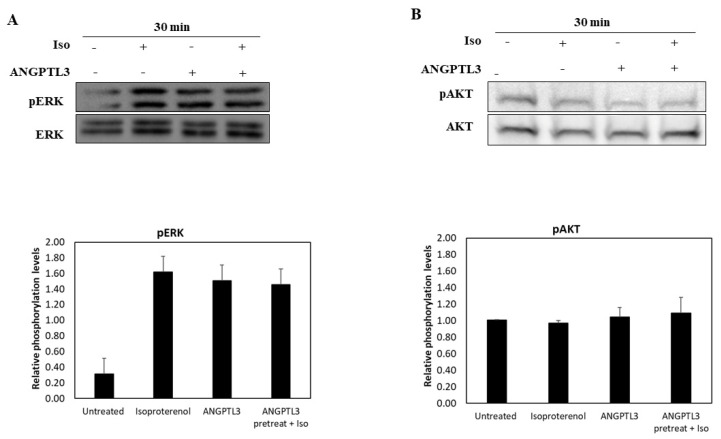
Intracellular kinase activation in ISO and ANGPTL3-treated cells. Representative blots of phosphorylated and total levels of ERK (**A**), AKT (**B**), in 3T3-L1 adipocytes treated for 30 min with recombinant human ANGPTL3 (ANGPTL3), ISO (Iso), and both. GAPDH levels were used as the loading control. Significant differences by *t*-test:

**Figure 3 biomolecules-12-00585-f003:**
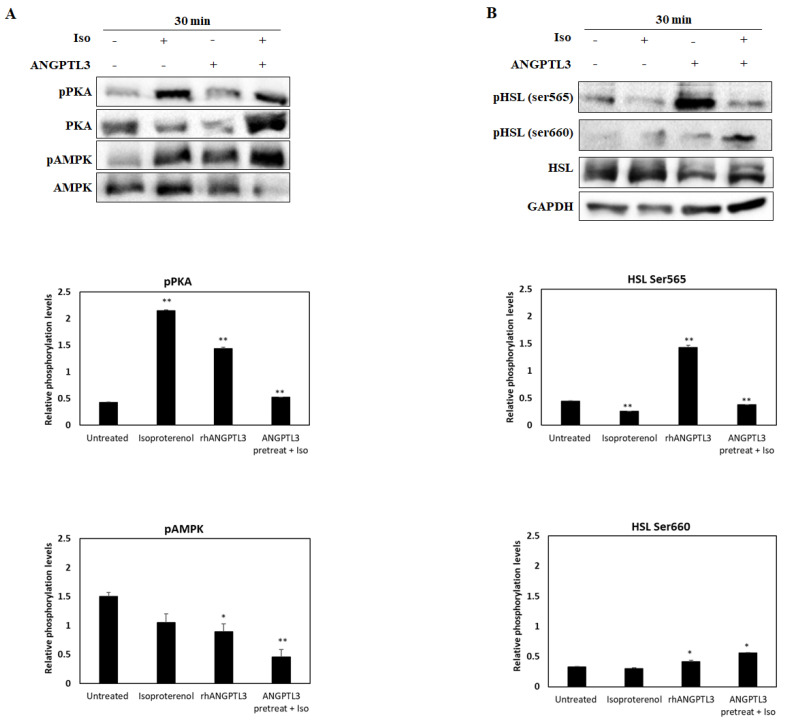
Lipolysis activation in ISO and ANGPTL3-treated cells. Representative blots of phosphorylated and total levels of AMPK and PKA in (Panel (**A**)). Representative blots of phosphorylated and total levels of HSL (Panel (**B**)) in 3T3-L1 adipocytes treated for 30 min with recombinant human ANGPTL3 (ANGPTL3), ISO (Iso), and both. GAPDH levels were used as the loading control. Multiple antibody hybridation. Significant differences by *t*-test: * *p* < 0.05, ** *p* < 0.01.

**Figure 4 biomolecules-12-00585-f004:**
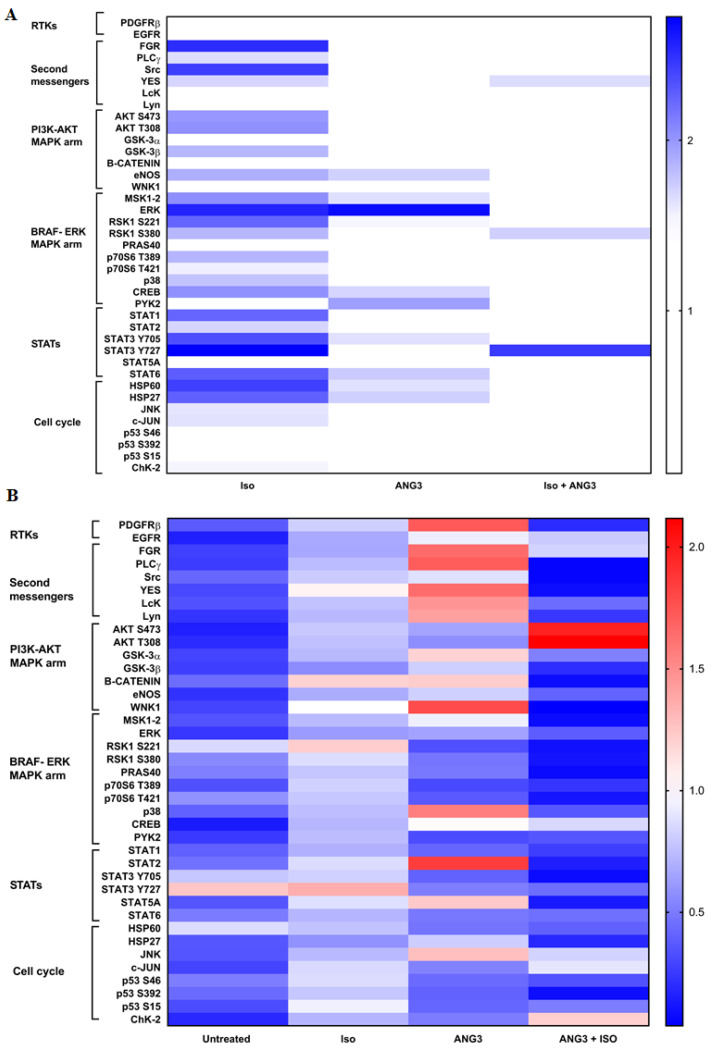
Proteome profiling heatmap—The y-axis reports the name of the 39 studied proteins in the proteome profiler assay, and the x-axis reports experimental conditions. (Panel (**A**)), the phosphorylation levels are normalized to the untreated sample and expressed as fold-change. Increased levels of phosphorylation are indicated by darker tones of blue, and the decreased levels are in white. (Panel (**B**)), phosphorylation levels after 48 h of vemurafenib inhibition; data are expressed as fold-change and normalized for the sample without inhibition. Increased levels of phosphorylation are indicated by darker tones of red, and the decreased levels are indicated by darker tones of red. ANG3—ANGPTL3-Fld treatment, ISO—ISO treatment, and ANG3 + ISO—combined treatment.

**Figure 5 biomolecules-12-00585-f005:**
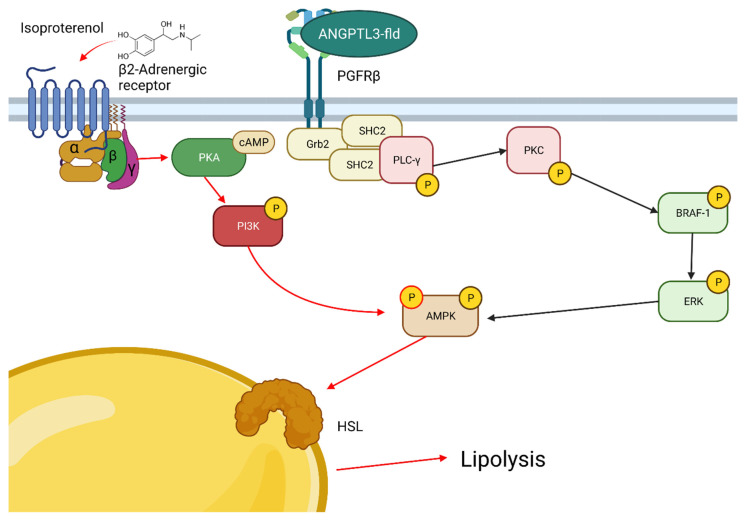
Molecular model describing the potential action ANGPTL3 fibrinogen-like domain function. Possible intracellular pathways activated by ANGPTL3fld and interaction with the β-adrenergic signaling. ANGPTL3-mediated molecular pathways are shown in black. β-adrenergic downstream pathways are shown in red.

## Data Availability

Data will be available on specific request to the corresponding authors.

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
