# Peer review of "The Fibrinogen-like Domain of ANGPTL3 Facilitates Lipolysis in 3T3-L1 Cells by Activating the Intracellular Erk Pathway"

_biomolecules, 2022, doi:10.3390/biom12040585_

Round 1

Reviewer 1 Report

Bini et al. based their study on the premise that the molecular mechanism of ANGPTL3-stimulated lipolysis in adipocytes remains unclear. Given that the C-terminal fibrinogen-like domain of ANGPTL3 (ANGPTL3-Fld) activates the AKT pathway in endothelial cells, they evaluated whether ANGPTL3-Fld stimulates lipolysis in adipocytes through a kinase pathway. Briefly, they treated 3T3-L1 adipocytes with isoproterenol (ISO), ANGPTL3-Fld, or combined, and evaluated lipolysis by releasing free fatty acids (FFA) to the medium. In addition, they assessed the activation status of intracellular kinases. As a result, they observed that ANGPTL3-Fld could not activate lipolysis, while the ANGPTL3-Fld/ISO combination increased FFA secretion two-fold compared to ISO alone. Furthermore, the ANGPTL3-Fld/ISO co-treatment resulted in HSL (hormone-sensitive lipase) activation. Finally, they postulate ANGPTL3-Fld as a facilitator of adipocyte lipolysis through a signal transduction pathway other than that mediated by the canonical β-adrenergic stimulus.
Although their findings contribute to the field, the authors must address the following concerns properly before publishing the study.
1. Abstract: Please restrict abbreviations to those accepted in common technical-scientific writing. Otherwise, it is hard to follow for readers with a non-specialized background.
2. Introduction: Although it contains a good scientific basis that supports the study, it is advisable to use succinct prose, as it is vague on several points. Furthermore, a thorough review of the writing style is suggested, and reader-based writing could improve reading flow.
3. Line 37: Please correct the text for LPP; it should be "lipoprotein lipase".
4. Lines 36-41: Please amend the abuse of references 1 and 2, and consolidate the data obtained-written from these information sources.
5. Lines 42-45: Please properly contrast the knowledge generated on the function of ANGPTL3 in mice (overexpression) and humans (deficiency), as the current writing is confusing.
6. Materials and Methods (lines 69-77, cell culture and treatments subsection): How did you ensure the cell differentiation of 3T3-L1 into adipocytes? Does the used method guarantee total differentiation? What biomarkers are used for cell discrimination? Can FACS be used?
7. Results and Discussion: The research methodology is standard. However, the scientific rigor is poor. Therefore, a thorough review of the experimental approach is advisable. In addition, scientific writing can be improved by following basic/common standards.
8. Lines 124-128: The interpretation of Fig. 1 can be improved if the experimental results are analyzed considering the mean rate of FFA secretion within the 30-60 min period (i.e., increase in uM FFA per min).
9. Lines 131-135: In Fig. 2, how can it be explained that pERK levels do not differ in the analyzed conditions? Do ISO and ANGPTL3-Fld comparatively stimulate the same signaling pathway? Furthermore, compete for the same receptor/protein involved in signaling? What will be the most likely explanation?
10. Figure 3, the GAPDH (control) panel is missing.
11. Finally, it is recommended to use the template provided by the journal and adhere to the format recommendations, as it facilitates the review process.
Given the above, the current manuscript version is not endorsed for publication.

Reviewer 2 Report

In the manuscript “The Fibrinogen-Like Domain Of Angptl3 Facilities Lipolysis In 3t3-L1 Cells By Activating The 3 Intracellular Erk Pathway” the authors attempt to elucidate the role of the fibrinogen-like domain of ANGPTL3 in lipolysis. Although the topic arises interest, I have few comments on the experimental design and the conclusion the authors reach:

  1. The kinase activation pattern is not consistent with figure 2 data: it is not clear if, in figure 4, ERK refers to pERK or total ERK levels. If it is the first case, pERK levels are not consistent with those described in figure 2. The same for total ERK, since its levels are not decreased in figure 2 while it seems very high after single treatments and low in cotreatment shown in figure 4A. How the authors explain this discrepancies?

  1. The authors state that AKT phosphorylation state in figure 2A is unchanged, but it seems that ANGPTL3-Fld decreases pAKT independently of ISO presence. I suggest the authors to show another representative western blot.

  1. How the authors explain reduced levels of pPKA upon ANGPTL3-Fld/ISO treatment with increased pAMPK and lipolysis? This point should be thoroughly discussed.

  1. It is quite puzzling how the ISO and ANGPTL3-Fld cotreatment does not resemble single treatments, and results depicted in figure 4B are even more confusing. This is the reason I suggest the authors to remove figure 4B. The conclusion that ANGPTL3-Fld has a ISO-dependent activity is reasonable, but how the authors explain the inability of Vemurafenib to completely shut down ISO/ANGPTL3-Fld pathway? In addition, the authors observed increased levels of PGDFRβ signal upon combined exposure to Vemurafenib and ANGPTL3-Fld, but how they exclude a non-specific or the irrelevancy of PDGFRβ-PANGPTL3-Fld axis to lipolysis?

  1. To answer these questions, I strongly suggest the authors to take advantage of genomic tools like shRNA. By silencing the activity of PKA, PLCγ and PDGFRβ, for instance, they should be able to determine which pathways are more relevant in mediating ANGPTL3-Fld activity.

Round 2

Reviewer 1 Report

The authors appropriately addressed the recommendations and modifications suggested during the initial review. Given this, the current version of the manuscript is endorsed for publication.

Reviewer 2 Report

I have no further comments.

Reviewer 3 Report

Albeit a number of the points criticized has been adequately addressed by the authors, not a single new experiment was added. Among them, the analysis of the effect of Vemurafenib on lipolysis seems to be of high relevance for the topic studied.  
